# Rootstock–Scion Exchanging mRNAs Participate in Watermelon Fruit Quality Improvement

**DOI:** 10.3390/ijms26115121

**Published:** 2025-05-27

**Authors:** Kang Ning, Weixin Zhou, Xiaoqi Cai, Leiyan Yan, Yuanchang Ma, An Xie, Yuhong Wang, Pei Xu

**Affiliations:** 1Key Laboratory of Specialty Agri-Product Quality and Hazard Controlling Technology of Zhejiang Province, College of Life Sciences, China Jiliang University, Hangzhou 310018, China; kning@cjlu.edu.cn (K.N.); s23090710082@cjlu.edu.cn (W.Z.); caixiaoqi2022@163.com (X.C.); p24091055043@cjlu.edu.cn (Y.M.); p23091055059@cjlu.edu.cn (A.X.); 2Ningbo Key Laboratory of Characteristic Horticultural Crops in Quality Adjustment and Resistance Breeding, Ningbo Academy of Agricultural Sciences, Ningbo 315042, China; lyyan1202@163.com

**Keywords:** mobile RNA, grafting, watermelon

## Abstract

Grafting significantly enhances plant quality, including stress resistance and fruit quality. We previously found that grafting watermelon onto pumpkin can alter the metabolite content, but the involvement of mobile RNA was unclear. Here, we established and comprehensively analyzed mobile mRNA (mb-mRNA) profiles, transcriptomes, and metabolomes between the rootstock (pumpkin) and scion (watermelon). A total of 834 mobile RNAs were identified in the pulp and stem of pumpkin-grafted watermelon. GO (Gene Ontology) and KO (Kyoto Encyclopedia of Genes and Genomes Orthology) analyses revealed photosynthesis- and carbon fixation-related mobile RNAs (e.g., Photosystem II D2, P700 chlorophyll a apoprotein) in the watermelon pulp and cell division-related mobile RNAs in the stem. Additionally, transcription factors like MADS and DNAJ exhibited mobility. The secondary structure prediction of the MADS-box transcription factor (*CmoCh20G002790*) showed multiple loop structures (e.g., internal and hairpin loops) related to its mobility. An integrated analysis of transcript and metabolite profiles indicated that photosynthesis-related products are regulated not only by the scion’s own RNA but also by mb-mRNA synthesized by the rootstock. This research advances our understanding of grafting’s molecular mechanisms and provides insights for improving crop quality and sustainability in agriculture.

## 1. Introduction

Grafting, a pivotal technique in artificial vegetative propagation, involves the fusion of a scion (a shoot or bud) from one plant with a rootstock (a stem or root system) from another, enabling their integration into a functional organism [1]. This method synergistically preserves desirable traits of scion cultivars—such as disease resistance and cold tolerance—while harnessing rootstock advantages like robust root systems and environmental resilience, ultimately enhancing plant productivity and stress adaptability. *Cucurbits* (watermelon, melon, and cucumber) and *Solanaceae* (tomatoes, peppers, and eggplant) are commonly and economically grafted. For instance, tomato grafting onto optimized rootstocks elevates yield, pathogen tolerance, and fruit quality [2,3]. A scion-to-rootstock-to-scion feedback signal loop in *cucurbits* improves the photosynthesis of the scion and further enhances production [4]. Grafting can also improve the drought resistance of pumpkins and the cold resistance of cucumbers [5,6]. For other horticultural crops, grafting can accumulate phenolic compound in grape berry skin [7]. Additionally, grafting facilitates dwarfing in woody species for size control and modulates secondary metabolites [8].

Despite its agricultural utility, the molecular underpinnings of grafting remain enigmatic. Successful graft union requires coordinated cellular reprogramming across three phases: tissue adhesion, callus proliferation, and vascular reconnection [9]. Key regulators include WOX (WUSCHEL-related homeobox) transcription factors. WOX13, rapidly induced post-wounding via AP2/ERF transcription factor WIND1, initiates a cascade by upregulating *WIND2* and *WIND3*, thereby driving cell dedifferentiation and organ regeneration. Concurrently, WOX13 activates cell wall-modifying enzymes (GLYCOSYL HYDROLASE 9Bs, PECTATE LYASE LIKEs, and EXPANSINs) to restructure cell walls, facilitating callus formation and vascular reconnection. Similarly, SlWOX4 serves as an early biomarker for vascular reconnection success in tomato grafts [10]. Notably, conserved molecular mechanisms exist across species: gymnosperm grafting studies identified *PHYTOCHROME A SIGNAL TRANSDUCTION 1 (PAT1)* as a core regulator, with shared expression patterns in injury response, cell division, and xylem formation genes [11]. The TOR (Target of Rapamycin) kinase integrates sugar and auxin signaling to promote graft healing [12], while environmental cues like far-red light (enhancing callus proliferation and vascular reconnection) and optimal temperatures (15–20 °C) critically regulate success rates [13]. Sugars act beyond energy provision, modulating cucumber–pumpkin graft healing via Glc-TOR signaling, with 5% glucose boosting survival [14].

The earliest evidence that biomacromolecules can be transferred between tissues came from plant viruses. The samples of phloem juice and grafting experiments further revealed mobile RNA molecules. Long-distance signaling between graft partners involves phloem-mobile molecules including DNA mRNAs, sRNAs, hormones, and metabolites [15]. The earliest evidence that biomacromolecules can be transferred between tissues came from plant viruses. The samples of phloem juice and grafting experiments further revealed mobile RNA molecules. The mobility of the FLOWERING LOCUS T (FT) gene is a classic example of mobile RNA in plants. The mRNA of FT, which is produced in the leaves, is transported over long distances to the shoot apical meristem, where it regulates flowering in plants, and this transport of TF can be achieved through grafting [16,17]. Stress-responsive mRNAs (e.g., pumpkin *HSP70* and low-temperature-induced mRNAs) mediate the adaptation of grafted plants to stress conditions [5,6]. Emerging sequencing technologies and bioinformatic pipelines [18] have accelerated discoveries of mobile RNAs. Non-RNA signals like γ-aminobutyric acid also contribute to long-distance communication [19].

Watermelon is a highly popular fruit, and grafting significantly impacts its growth and yield. This has led to the widespread adoption of grafting in watermelon production. In a previous study, we used the transcriptome and metabolome to reveal that watermelon grafted onto pumpkin improves watermelon fruit quality and changes genes in the stem cell walls [20]. In this study, we further explore the effects of mb-mRNAs on grafted watermelon. Totally, we identified 834 mobile RNAs in pumpkin-grafted watermelon. A functional analysis showed that photosynthesis- and carbon fixation-related mobile RNAs were identified in the watermelon pulp, and cell division-related mobile RNAs were identified in the stem. Multiple loop structures were found in the MADS-box transcription factor (CmoCh20G002790). An integrated analysis of transcript and metabolite profiles showed that photosynthesis-related products are regulated by both the scion RNAs and mobile RNAs. Our study advances our understanding of grafting’s molecular mechanisms and provides insights for improving crop quality.

## 2. Results and Discussion

### 2.1. Identification of mb-mRNAs

We identify putative mb-mRNAs from the RNA-Seq data according to the pipeline [18]. Deep coverage sequencing was performed, and a total of 58,043,296 to 495,488,346 reads were identified in the samples by RNA-Seq. The reads were first mapped to the watermelon genome, with 84.62% to 96.49% of the reads being mapped. In the samples XG, XJ, YG, and YJ, 10,020,112, 4,963,562, 69,466,057, and 17,008,725 unmapped reads were identified, respectively. These unmapped reads are candidates for infiltrating reads. These reads were then mapped to the pumpkin genome, and a total of 32,064 to 495,277 reads (0.36% to 3.05%) were mapped to the pumpkin genome (Table 1, Appendix A). Our results suggest that the mobility of RNA is related to distance: more pumpkin mobile RNA (mb-mRNA) was identified in the stem, which is closer to the rootstock, than in the fruit, which is farther from the rootstock. This is similar to one view of the long-distance transport of mb-mRNA, which posits that over a sufficiently long period of time, the mRNA produced by the rootstock can be transferred from cell to cell [4]. A total of 834 genes were identified as mb-mRNAs, indicating that a large number of mRNAs move directly between the scion and rootstock during grafting.

### 2.2. Validation of mb-mRNAs

To confirm the reliability of the mb-mRNA-seq data, Pre-mRNA splicing factor SYF1 (*CmoCh03G005050*), Photosystem II protein D1 (*CmoCh18G007080*), YMF19 domain-containing protein (*CmoCh12G007630*), and Heat shock protein 90 (*CmoCh02G002280*) mb-mRNAs were selected and RT-PCR analyses were performed. The mb-mRNA-seq results showed that these genes were highly expressed in the stem of pumpkin-grafted watermelon, and low levels of mRNA were detected in the pulp of pumpkin-grafted watermelon (Figure 1). This suggests that these genes may have moved from the pumpkin rootstock to the watermelon fruit. The RT-PCR data showed a similar expression pattern, which suggests that our mb-mRNAs data are reliable.

### 2.3. Identification of Putative mb-mRNAs

We first identified the mb-RNAs that had translocated from the pumpkin rootstock to the cucumber scion. In total, 34 and 27 mb-RNAs were identified in YJ vs. XJ and YG vs. XG, respectively (Appendix A). Six genes were detected in both comparisons. We then performed GO and KEGG analyses of these mb-mRNAs (Figure 2). The mb-mRNAs identified in the YG vs. XG comparison were mainly enriched in pathways such as 2-oxocarboxylic acid metabolism, photosynthesis, and arachidonic acid metabolism. In the Biological Process (BP) category, these genes were primarily enriched in protein metabolic processes, organonitrogen compound metabolic processes, metabolic processes, and primary metabolic processes. In the Molecular Function (MF) group, the most significant GO terms included oxidoreductase activity and metal ion binding. These genes were also enriched in cellular anatomical entities and intracellular organelles in the Cellular Component (CC) category. Previous studies have found that grafting can increase photosynthesis in scions [4]. In our study, several genes involved in photosynthesis were detected in the pumpkin-grafted watermelon, including Photosystem II D2 protein (*CmoCh10G011100*), NADH dehydrogenase subunit 7 (*CmoCh12G007650*), Photosystem I P700 chlorophyll an apoprotein A1 (*CmoCh00G000560*), and NAD(P)H-quinone oxidoreductase subunit 2 (*CmoCh00G001260*). The enrichment of these RNAs will translate more proteins, thereby affecting the quality of grafted watermelon fruits.

The mb-mRNAs identified in the YJ vs. XJ comparison were mainly enriched in pathways such as autophagy, propanoate metabolism, spliceosome, and aminoacyl-tRNA biosynthesis. In the Molecular Function (MF) category, these genes were primarily enriched in RNA binding, nucleic acid binding, and heterocyclic compound binding. The most significant GO terms in the Biological Process (BP) group included transport, cellular process, and the establishment of localization. In the Cellular Component (CC) category, genes were mainly enriched in the cytoplasm and cellular anatomical entity. RNA-seq showed that cell division related genes were highly expressed in the stem of the grafted watermelon [20]. Mb-RNAs like Small Polypeptide DEVIL 11 (*CmoCh04G012470*), Protein Translation Factor SUI1 Homolog (*CmoCh19G009550*), Ferredoxin-NADP Reductase (*CmoCh03G014440*), BTB/POZ Domain-Containing Protein TNFAIP1 (*CmoCh20G010290*), and BSD Domain-Containing Protein (*CmoCh18G011100*) were detected in the stem of pumpkin-grafted watermelon. Mobile RNAs in fruits are involved in metabolite synthesis. Our previous study found that grafting can affect watermelon fruit quality, and mb-mRNAs may also participate in these changes. The pumpkin mb-mRNAs identified in the stem have functions related to transport and binding, indicating that these mb-mRNAs might alter the transport functions of the stem and, consequently, affect the quality of watermelon fruit. mb-RNA have been identified in several species, including apple, cucumber, and grape [21,22]. Although previous studies have explored the mb-RNAs produced in pumpkin rootstocks that can move into watermelon scions, our data identified different mb-RNAs, further contributing to the understanding of how pumpkin rootstocks improve scion quality. Mb-RNA-seq only identified some mb-RNAs that moved from the rootstock to the scion. However, it is not clear whether the translation efficiency of these mobile RNAs is similar to that of scion-produced RNAs. In the future, the identification of the proteins produced by rootstocks in scions by proteomics technology might fully prove that mobile RNAs are involved in the improvement of the quality of scion fruits.

### 2.4. Identification of Specific mb-mRNAs by Mfuzz

To identify specific mb-mRNAs in the pulp and stem of pumpkin-grafted watermelon, we performed an Mfuzz time-course analysis of the mb-mRNA-seq data. Ultimately, five gene clusters were obtained. Genes belonging to Class 1, which contain 208 mb-mRNAs, showed high expression in YG, while genes in Class 5, which contain 411 mb-mRNAs, showed high expression in YJ. Genes belonging to cluster 2 are highly expressed in both YG and YJ (Figure 3a). 

We then performed a functional analysis of these specific mb-mRNAs. Studies have shown that grafting can change the metabolite content in the rootstock, including carbohydrate, lipid, fatty acid, etc. In this study, mb-mRNAs that were specifically detected in YG (Class 1) are enriched in pathways such as metabolite biosynthesis. For example, Phosphoglucomutase (*CmoCh04G022160*) catalyzes the reversible conversion of glucose-1-phosphate to glucose-6-phosphate, a key step in starch metabolism and sugar mobilization [23], which is important for energy metabolism and carbon allocation in plants. Cytochrome b5-like proteins (*CmoCh14G019410*) are heme-binding proteins involved in electron transfer reactions. In plants, they play roles in processes such as fatty acid desaturation, sterol biosynthesis, and detoxification. Hexosyltransferases (*CmoCh09G008080*, *CmoCh17G000010*) are involved in the synthesis of glycosides by transferring hexose sugars to various acceptors. In plants, these enzymes contribute to the biosynthesis of cell wall components, glycoproteins, and secondary metabolites [24]. In previous studies, we found that grafting can improve watermelon fruit quality [20]. The presence of these mb-mRNAs in watermelon fruits may contribute to the synthesis of certain substances, which could be responsible for the changes in watermelon quality caused by grafting. Similarly to the previous studies [4], several photosynthesis-related genes, including Photosystem I P700 chlorophyll a apoprotein A1 (*CmoCh00G000560*), NADH dehydrogenase subunit 7 (*CmoCh12G007650*), and Photosystem II D2 protein (*CmoCh10G011100*), were identified in scions, indicating that grafting could increase photosynthesis in the aboveground section (Figure 3b).

Signal transduction between the rootstock and scion plays a key role in plant grafting, which not only affects grafting healing and plant growth and development, but also has an important impact on plant stress resistance and quality. Mb-RNA, as a signal, long-distance transport, and regulation effect, affect the growth and development of grafted plants, phenotypic changes, stress resistance, and other important biological processes, such as ADP-ribosylation factors (ARFs) (*CmoCh17G009890*), which are small GTP-binding proteins involved in vesicle trafficking and membrane dynamics. In plants, ARFs play a vital role in processes like cell wall formation, intracellular transport, endocytosis, and exocytosis, which are critical for growth and response to environmental stimuli. Mo25 proteins (*CmoCh13G004010*) function as scaffolding proteins that stabilize kinase complexes. In plants, they are associated with stress signaling, growth regulation, and possibly nutrient homeostasis [25].

Transcription factors can also move from the rootstock to the scion. Several transcriptional factors were also highly expressed in YG. For example, DNAJ proteins are important chaperones that maintain the stability of protein complexes in plant cells [26]. mRNA of pumpkin DNAJ (*CmoCh03G012830*) were moved to the fruit of the scion. The MADS family is not only involved in fruit ripening, but also affects fruit quality, such as regulating the expression of sugar metabolism genes, affecting fruit sugar content and flavor, and also affecting fruit firmness. Pumpkin MADS (*CmoCh20G002790*) was also detected in the watermelon. PHD Finger (*CmoCh05G004300*) is involved in the remodeling and modification of chromatin, interacting with histone acetyltransferase (HAT) or histone demethylase to influence the open or closed state of chromatin. These transcription factors transport form the rootstock to contribute to the watermelon development.

While several mb-mRNAs involved in cell development were identified in YJ (Class 5), their functions are diverse and crucial for various cellular processes. For example, the Ubiquitin Receptor RAD23 is involved in processes like hormone signaling, stress tolerance, and cellular homeostasis in plants [27]. Glycine-rich proteins are structural proteins often associated with the cell wall [28]. They play roles in strengthening cell walls, facilitating plant growth, and responding to environmental stresses such as pathogen attacks, cold, and drought. Mannan Endo-1, 4-beta-Mannosidase, also known as beta-mannanase, catalyzes the hydrolysis of mannans and glucomannans, which are components of hemicellulose in plant cell walls. It is involved in cell wall remodeling, seed germination (by breaking down cell wall polysaccharides in the endosperm), and fruit ripening processes. E6-like proteins are typically associated with cell proliferation and differentiation. In plants, they may regulate growth and reproductive development or contribute to stress responses by controlling cell division and expansion. The transcription factor E2FB (*CmoCh20G001680*) belongs to the E2F/DP family, which regulates the cell cycle by controlling the expression of genes involved in DNA replication and cell division. In plants, E2FB is essential for meristem activity, leaf growth, and responding to external stimuli by promoting or repressing cell proliferation. Homologous pumpkin genes were detected in the stem of the hetero-grafts, suggesting that mb-mRNAs were involved in the healing process between the rootstock and scion (Figure 3b).

These results show that mobile RNAs can move into the pulp of watermelon to alter fruit quality, while mobile RNAs in stem participates in cell division and promotes graft healing. Previously, our transcriptome data showed that genes related to cell division and proliferation were highly expressed in the stem of the pumpkin-grafted watermelon. The RNA produced by the scion and the mobile RNAs produced by the rootstock jointly regulate the growth and development of plants and fruit quality.

### 2.5. The Structure of mb-mRNA

Studies have suggested that RNA motifs are important factors in their ability to move, such as Polypyrimidine (poly-CU) sequences and transfer RNA (tRNA)-related sequences. RNA secondary structures, including tRNA-like structures (TLS), the untranslated regions (UTRs) of mRNAs, and the stem-loop structures of pre-miRNAs, also contribute to RNA mobility [15,29]. RNA methylation (methylated 5′ cytosine, m5C) contributes to RNA transport and function as well [30].

To detect whether the mb-RNA identified in watermelon pulp contained loop secondary structures, a MADS-box transcription factor (CmoCh20G002790) was selected. By using the RNAfold web server, multiple loops were identified in the sequence of CmoCh20G002790, including external loops, internal loops, and hairpin loops (Figure 4a, Appendix A). Internal loops are typically located within the RNA structure, between stem regions, and can serve as binding sites for proteins or other molecules, thereby influencing RNA movement efficiency [31]. Over 300 internal loops were identified in the CmoCh20G002790 sequence. These loops might contribute partially to the RNA’s long-distance movement. Additionally, the entropy value at the 5′ end of CmoCh20G002790 is higher than that in the middle region, suggesting greater variability at the 5′ end compared to the middle region (Figure 4b), suggesting that the 5′ end might be responsible for the mobility of the RNA of *CmoCh20G002790*.

### 2.6. Integrated Analysis of Transcript and Metabolite Profiles

To further reveal the effects of RNAs and mobile RNAs on grafted watermelon quality, we combined transcriptome, movable transcriptome, and metabolome data for analysis. We focused on different accumulated metabolites, different expressed genes, and different expressed mobile RNAs in the YG vs. XG comparison. Photosynthesis, starch and sucrose metabolism, and carbon metabolism pathways genes and metabolites showed different accumulation between YG and XG (Figure 5a–c). Several genes identified in the RNA-seq were validated by RT-PCR (Appendix A). We then calculated the correlation between metabolites and RNA/mobile RNAs, and the results showed that metabolites are regulated by both RNA and mb-RNA. For example, D-Sedoheptulose 7-phosphate is a key intermediate in the metabolism of plants, particularly in the sugar phosphate pathways and photosynthesis. RNA of pfkB-like carbohydrate kinase family protein and Trehalose 6-phosphate phosphatase were highly positive correlated with D-Sedoheptulose 7-phosphate. Meanwhile, Photosystem II D2 protein mobile-RNA were positively correlated with D-Sedoheptulose 7-phosphate. These carbohydrates produced during photosynthesis can provide precursor substances for other carbohydrates, fatty acids, and secondary metabolites, thereby improving the quality of the draft watermelon. Grafting can change photosynthesis and carbon fixation in watermelon scions, and genes in scions may play an important role in this process, while mobile RNA in the rootstock may also participate in related biological processes, but the participation may be small, because the content of mobile RNA is less than that produced by scions. More in-depth research on these mb-RNAs requires biotechnologies such as gene editing.

Pumpkin rootstocks are widely used in grafting, mainly due to their strong vitality, including enhancing photosynthesis and improving stress resistance [5,32,33,34,35]. Our results show that pumpkin rootstocks produce mb-RNA related to photosynthesis and carbon fixation. Liu et al. also found that pumpkin rootstocks can donate many mb-RNAs, which participate in carbon fixation and oxidative and chlorophyll metabolism in the cucumber scion [4]. Hormone signal transduction-related mb-RNAs can move from the rootstock to scion. It was also found that RNAs that contributed to the biosynthesis of amino acids and secondary metabolites can move from the scion to the rootstock [36,37]. In the future, our experimental material (watermelon grafted onto pumpkin) can be used to detect the mb-RNAs produced by scions present in the rootstock, which might reveal the influence of the scion on the rootstock. As a whole, the rootstock of the plant will produce mb-RNAs to improve the fruit quality of the scion. Conversely, the scion may also produce mb-RNAs to promote the development of the rootstock’s root system, ultimately enabling the entire grafted plant to grow better.

## 3. Materials and Methods

### 3.1. Identification of mb-mRNAs by RNA-Seq

Total RNA was isolated using the Quick RNA Isolation Kit (Waryoung, Beijing, China), with RNA-seq procedures following our established protocol [13]. In brief, the fresh samples were taken from the plants and frozen in liquid nitrogen. The samples were ground into powder under the liquid nitrogen. A cell lysis buffer was added into the powder and the total RNAs were adsorbed onto the colon column; after protein digestion and washing, the total RNA was obtained.

For RNA-seq, the method was as follows: 1 μg RNA was used to preform RNA-seq transcriptome libraries. The TruSeqTM RNA sample preparation kit was used for fragmentation, cDNA synthesis, and ligation adaptors. Paired-end libraries were sequenced by Illumina NovaSeq6000 sequencing (Shanghai BIOZERON Co., Ltd., Shanghai, China).

For data analysis, the clean reads were separately aligned to the reference genome with an orientation mode using hisat2 (https://ccb.jhu.edu/software/hisat2/index.shtml, accessed on 6 August 2023) software. Mb-mRNA identification was performed according to the computational pipeline described by Wang et al. (2020) [18]. Briefly, the raw paired end reads were trimmed and quality controlled by Trimmomatic with parameters (SLIDINGWINDOW: 4:15 MINLEN:75) (version 0.36. http://www.usadellab.org/cms/uploads/supplementary/Trimmomatic, accessed on 8 August 2023). Paired-end reads from transcriptome sequencing were initially aligned to the watermelon reference genome (ASM3963971v1; NCBI accession GCA_039639715.1) using HISAT2. Unmapped reads were subsequently subjected to BlastN analysis against the pumpkin genome (Cmos 1.0; NCBI accession GCF_002738365.1). Reads demonstrating ≥95% sequence identity to pumpkin genomic regions were classified as candidate mb-mRNAs. To eliminate false positives, we excluded genes detected in homograft controls (watermelon scions/watermelon rootstocks) that showed cross-mapping to the pumpkin genome. Different expressed mb-RNAs were identified as |logFC| ≥ 1 and displayed false discovery rates (FDRs) ≤ 0.05.

### 3.2. RT-PCR

Selected candidate mobile RNAs were validated by reverse transcription PCR. Total RNA was extracted as described above, and cDNA synthesis was performed using the T7 RNAi Transcription Kit (Vazyme, Nanjing, China). The pumpkin gene *CmoRUL* (CmoCh15G009810) served as the reference control [14], and a-Tubulin was used as the watermelon reference gene [20]. Relative expression levels were calculated via the 2^−ΔΔCt^ method [15], with three technical replicates analyzed per biological sample. Primer sequences are provided in Appendix A.

### 3.3. Bioinformatics Analysis

Mfuzz time-course analysis was performed using the Metware Cloud (https://cloud.metware.cn, accessed on 8 January 2025). Heatmaps of all the mobile RNAs identified in this study were drawn by OmicStudio (https://www.omicstudio.cn/tool, accessed on 8 January 2025). The RNA structure was predicted by online tools (https://rna.urmc.rochester.edu/RNAstructureWeb/Servers/Fold/Fold.html, accessed on 8 January 2025).

Based on the content data of RNAs, mb-RNAs, and metabolites in four samples (XG, YG, XJ, and YJ), the spearman correlation between each RNA, mb-RNA, and metabolite was calculated using the ggplot2 (3.35). Then, based on the spearman of each RNA, mb-RNA, and metabolite, the correlation network was drawn by igraph (1.2.6). The correlation analysis and correlation network was achieved using the OmicStudio tools at https://www.omicstudio.cn/tool/ (accessed on 8 January 2025).

### 3.4. RNA Secondary Structure Prediction

The secondary structure of the MADS-box transcription factor (CmoCh20G002790) was modeled using the RNAfold web server (http://rna.tbi.univie.ac.at/cgi-bin/RNAWebSuite/RNAfold.cgi, accessed on 8 February 2025).

## 4. Conclusions

In this study, we comprehensively investigated the identification, validation, and functional analysis of mb-mRNAs in grafted watermelon plants. A total of 834 genes were identified as mb-mRNAs, indicating significant RNA mobility between the pumpkin rootstock and the watermelon scion. The validation of selected mb-mRNAs via RT-PCR confirmed the reliability of the sequencing data. Functional analyses revealed that these mb-mRNAs are involved in various biological processes, including photosynthesis, metabolism, and cell development, suggesting that they play crucial roles in altering fruit quality and promoting graft healing. Additionally, we analyzed the RNA secondary structure of the MADS transcription factor (*CmoCh20G002790*), and multiple loops were found, which might be necessary for its mobility. The integrated analysis demonstrated that both RNA and mb-RNA contribute to the regulation of metabolite accumulation (starch and sucrose metabolism and carbon metabolism) in grafted plants. In the future, the movement mechanism of mobile RNA can be explored, and ultimately the quality of watermelons can be improved by increasing the amount of mobile RNA.

## Figures and Tables

**Figure 1 ijms-26-05121-f001:**
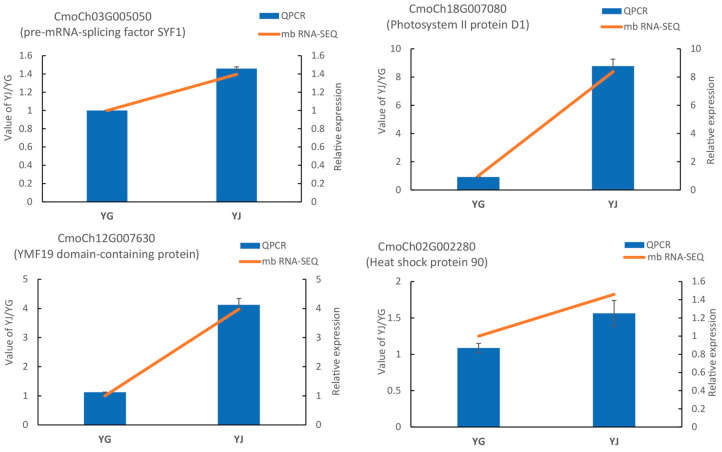
RT-PCR of the mb-mRNAs to validate data reliability. RT-PCR analysis and mb-mRNAs data of 4 genes (*CmoCh03G005050*, *CmoCh18G007080*, *CmoCh12G007630,* and *CmoCh02G002280*). The RT-PCR data are shown in the blue column, while the mb-mRNAs data are represented by the orange line.

**Figure 2 ijms-26-05121-f002:**
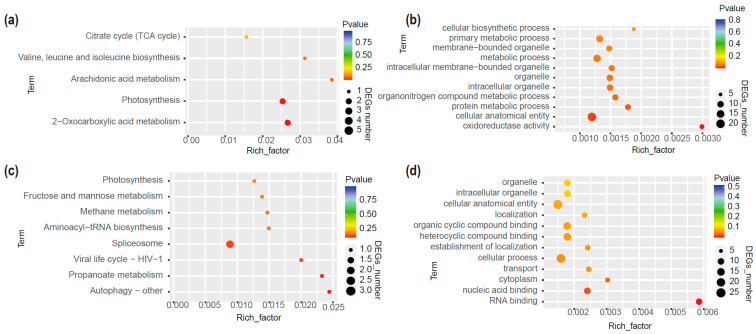
Functional analysis of the mobile RNAs identified in different grafted plants. (**a**) KO (Kyoto Encyclopedia of Genes and Genomes Orthology) analysis of mobile RNAs identified in YG vs. XG. (**b**) GO (Gene Ontology) analysis of mobile RNAs identified in YG vs. XG. (**c**) KO analysis of mobile RNAs identified in YJ vs. XJ. (**d**) GO analysis of mobile RNAs identified in YJ vs. XJ.

**Figure 3 ijms-26-05121-f003:**
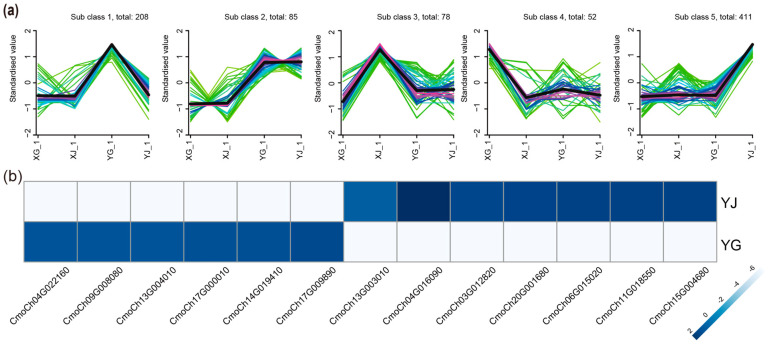
Organ-specific Mb-mRNA function analysis. (**a**) Mfuzz time-course analysis of mb-mRNAs. Five subclasses were obtained through Mfuzz clustering. The log_2_ (FPKM) values of the mb-mRNAs in each sample were calculated, and the expression patterns of each mb-mRNA were plotted over time. Each gene is represented by a green line, with overlapping genes showing a darker color. The black line indicates the overall trend of the gene set. (**b**) Heatmap of selected mb-mRNAs. The log_2_ (FPKM) values of the mb-mRNAs in each sample were calculated. In the heatmap, blue indicates high expression, while white indicates low expression.

**Figure 4 ijms-26-05121-f004:**
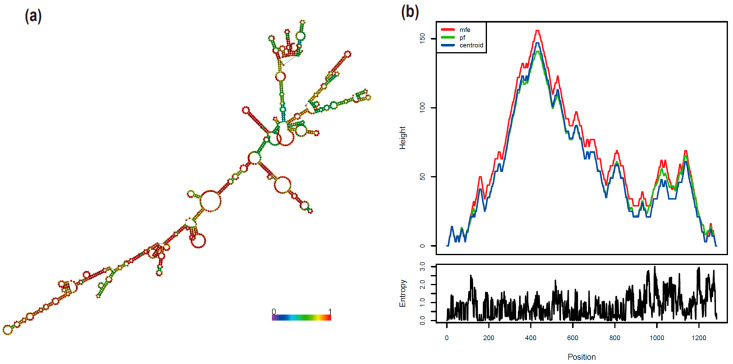
Computer-predicted secondary structure of MADS-box transcription factor (CmoCh20G002790). (**a**) Graphical structure of CmoCh20G002790 RNA. The structure is colored by base-pairing probabilities. For unpaired regions, the color denotes the probability of being unpaired. (**b**) Mountain plot representation of the CmoCh20G002790 RNA structure, the thermodynamic ensemble of RNA structures, and the centroid structure. Red line (MFE) represents the minimum free energy structure of the RNA. Green line (pf) represents the partition function, which considers the thermodynamic ensemble of all possible RNA structures. Blue line (centroid) represents the centroid structure, which is the structure closest to all other structures in the ensemble.

**Figure 5 ijms-26-05121-f005:**
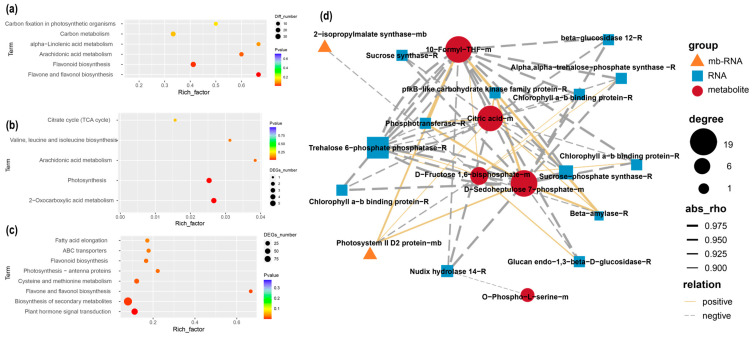
Integrated analysis of transcript and metabolite profiles. KO analysis of different accumulated metabolites (**a**), different expressed mobile genes (**b**), and different expressed genes (**c**) between YG and XG. (**d**) Networks involving the metabolites, RNA, and mb-RNA that are involved in photosynthesis and carbon metabolism pathways. Red circles indicate the metabolites, blue quadrate indicate the RNAs and orange triangle indicate the mobile RNAs.

**Table 1 ijms-26-05121-t001:** Summary of mobile RNA-seq data.

SampleID	Pumpkin Alignment			Watermelon Alignment		
	Clean Reads	Mapped Reads	Mapped Rate (%)	Clean Reads	Mapped Reads	Mapped Rate (%)
XG_1	8,828,672	32,064	0.36	60,528,066	51,699,394	85.41
XG_2	9,883,596	37,851	0.38	65,356,312	55,472,716	84.88
XG_3	11,348,068	53,431	0.47	72,865,870	61,517,802	84.43
XJ_1	5,499,504	47,294	0.86	91,199,184	85,699,680	93.97
XJ_2	5,855,008	46,574	0.8	84,643,282	78,788,274	93.08
XJ_3	3,536,176	32,162	0.91	58,043,296	54,507,120	93.91
YG_1	67,836,182	244,339	0.36	450,405,130	382,568,948	84.94
YG_2	71,149,316	270,573	0.38	462,696,550	391,547,234	84.62
YG_3	69,412,674	276,379	0.4	453,891,800	384,479,126	84.71
YJ_1	16,256,512	495,277	3.05	462,670,224	446,413,712	96.49
YJ_2	15,682,756	241,877	1.54	418,630,324	402,947,568	96.25
YJ_3	19,086,908	315,904	1.66	495,488,346	476,401,438	96.15

## Data Availability

Data will be made available on request. The raw RNA-seq data. (Accession no. PRJNA1151145) were uploaded to NCBI.

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
