# Peer review of "Rootstock–Scion Exchanging mRNAs Participate in Watermelon Fruit Quality Improvement"

_ijms, 2025, doi:10.3390/ijms26115121_

Round 1
Reviewer 1 Report
Comments and Suggestions for Authors
A review of the manuscript entitled: Rootstock-scion exchanging mRNAs participate in watermelon fruit quality improvement.
The research presented in this manuscript represents a topic of significant scientific and applied importance. The authors address the current problem of mobile mRNAs (mb-mRNAs) role in regulating fruit quality in grafted plants, focusing on the model watermelon/pumpkin system. Integrating transcriptomics, bioinformatics, RNA structure prediction, and metabolomics characterizes the study's interdisciplinary nature. The experimental approach, which is founded on RNA-seq analysis and RT-PCR verification of selected genes, is regarded as robust.
Notwithstanding the introduction of new data concerning the number and characteristics of mb-mRNAs, certain aspects require critical revision. The manuscript presents an analysis of the current state of experimental evidence surrounding the functionality of mb-mRNAs (i.e., whether mRNA transcribed from the rootstock is translated in the scion). Conclusions regarding the effect of mb-mRNAs on fruit quality are based on correlation rather than direct functional evidence (e.g., overexpression or mutant experiments are lacking). The spatial data are lacking: the authors do not present transcript localisation or perform in situ hybridisation or translational reporters. The manuscript lacks statistical validation of the observed differences in mb-mRNA expression. While the numbers and observations are presented, the statistical significance of these differences remains ambiguous. In the integrated transcriptome-metabolome analysis, the methodology for calculating correlations is not transparent, as correlation coefficients and significance levels are absent.
The discussion is intriguing. However, I think it would benefit from further elaboration. The authors hypothesise that mobile mRNAs regulate fruit quality; however, it is unclear whether the mere presence of mRNAs results in protein synthesis or metabolic regulation. I think it's a good idea to exercise caution when formulating conclusions.The limitations of the RNA-seq technique in the context of mobile RNAs should be emphasised, as the lack of evidence for translation (e.g., by proteomics) is a significant shortcoming.
Also, no discussion about potential selection mechanisms for mobile RNAs is provided. While the authors refer to secondary structures, there is a lack of indication as to which motifs are essential and whether mutational data supports their function.
In conclusion, the paper brings interesting observations on mRNA mobility and its potential role in fruit quality but needs more convincing documentation of functional mechanisms.
Comments on the Quality of English LanguageDespite the generally understandable language, the manuscript contains numerous stylistic and grammatical errors.
Author Response
A review of the manuscript entitled: Rootstock-scion exchanging mRNAs participate in watermelon fruit quality improvement. The research presented in this manuscript represents a topic of significant scientific and applied importance.
The authors address the current problem of mobile mRNAs (mb-mRNAs) role in regulating fruit quality in grafted plants, focusing on the model watermelon/pumpkin system. Integrating transcriptomics, bioinformatics, RNA structure prediction, and metabolomics characterizes the study's interdisciplinary nature. The experimental approach, which is founded on RNA-seq analysis and RT-PCR verification of selected genes, is regarded as robust.
Notwithstanding the introduction of new data concerning the number and characteristics of mb-mRNAs, certain aspects require critical revision. The manuscript presents an analysis of the current state of experimental evidence surrounding the functionality of mb-mRNAs (i.e., whether mRNA transcribed from the rootstock is translated in the scion). Conclusions regarding the effect of mb-mRNAs on fruit quality are based on correlation rather than direct functional evidence (e.g., overexpression or mutant experiments are lacking). The spatial data are lacking: the authors do not present transcript localization or perform in situ hybridization or translational reporters.
Response: A very good suggestion. Indeed, spatial data can significantly enhance the reliability of this study. Based on omics data (transcriptome, metabolome, and mobile RNA-seq), this study comprehensively analyzed the three omics datasets, revealing that RNAs and mobile RNAs jointly influence the quality of grafted watermelon. However, these conclusions are primarily based on the omics data, and our study lacks experiments such as in situ hybridization. But we think that samples (pulp and stem) of our omics can showed the localization of the RNAs and mobile RNAs. Additionally, the experiment was conducted in 2023, and currently, there are no related samples available for in situ hybridization.
The manuscript lacks statistical validation of the observed differences in mb-mRNA expression. While the numbers and observations are presented, the statistical significance of these differences remains ambiguous. In the integrated transcriptome-metabolome analysis, the methodology for calculating correlations is not transparent, as correlation coefficients and significance levels are absent.
Response: we have added the identification of different expressed mb-mRNAs in the ‘Materials and Methods- Identification of mb-mRNAs by RNA-seq’. We also have added the Correlation analysis and correlation Network protocol in the ‘Materials and Methods- Bioinformatics Analysis’.
The discussion is intriguing. However, I think it would benefit from further elaboration. The authors hypothesize that mobile mRNAs regulate fruit quality; however, it is unclear whether the mere presence of mRNAs results in protein synthesis or metabolic regulation. I think it's a good idea to exercise caution when formulating conclusions. The limitations of the RNA-seq technique in the context of mobile RNAs should be emphasized, as the lack of evidence for translation (e.g., by proteomics) is a significant shortcoming. Also, no discussion about potential selection mechanisms for mobile RNAs is provided. While the authors refer to secondary structures, there is a lack of indication as to which motifs are essential and whether mutational data supports their function.
Response: That’s a good suggestion. We have added discussions on these contents in the discussion section. Due to the lack of mutants, we cannot accurately determine the structures that significantly impact RNA mobility. According to the literature, “The sequences of Arabidopsis GA-INSENSITIVE RNA constitute the motifs that are necessary and sufficient for RNA long-distance trafficking.” However, based on predictions from the computer program MFOLD, we believe that structures with high entropy values have a more significant impact on RNA mobility. We have included these findings in the manuscript.
In conclusion, the paper brings interesting observations on mRNA mobility and its potential role in fruit quality but needs more convincing documentation of functional mechanisms.
Response: We have modified the conclusion section, added specific genetic information, and prospected the future research directions.
Despite the generally understandable language, the manuscript contains numerous stylistic and grammatical errors.
Response: We have changed the errors in the manuscript.

Reviewer 2 Report
Comments and Suggestions for Authors
Comment to authors:
The manuscript established and comprehensively analyzed mobile mRNA profiles, transcriptomes, and metabolomes between rootstock (pumpkin) and scion (watermelon). The result indicated that photosynthesis-related products were regulated not only by the scion's own RNA but also by mobile mRNA from the rootstock.
However, the manuscript is completely based on the sequencing data analysis of RNA-seq and metabolomes without any verification. After careful reading and evaluation, there are major problems in the manuscript at current state. I recommend rejection of the manuscript. The specific comments are as follows:
- The article lacks innovation. Although the study compared the mobile mRNA profiles, transcriptomes, and metabolomes from the rootstock (pumpkin) and scion (watermelon), the methodology of the study lacked sufficient innovation. The findings of this paper are similar to those of the existing literature and failed to provide new insights or breakthrough discoveries.
- The authors' interpretation of the results in the Discussion was superficial. Several findings are presented as descriptive data without mechanistic explanations. For instance, how do photosynthesis-related mb-mRNAs specifically regulate metabolite synthesis? It is recommended to enhance the Discussion by linking these observations to signaling pathways or established models.
- The quality of graphs and charts needs to be improved. Using clearer labeling and color differentiation.
- The description of RNA extraction and sequencing protocols is overly brief. Critical parameters must be added to enhance reproducibility, including: Sequencing depth, Alignment software version and specific parameter settings, Filtering thresholds for raw reads.
- The terminology needs to be harmonized: “mb-mRNA” and “mobile mRNA” are used interchangeably in the text, and it is suggested that the term “mb-mRNA” be harmonized throughout the text.
- The current conclusion is overly general. It is recommended to explicitly summarize key findings, such as the roles of specific genes (e.g., CmoCh20G002790 [MADS-box transcription factor] or pathways (e.g., photosynthesis and carbon fixation), and to clearly outline future research directions.
Author Response
The manuscript established and comprehensively analyzed mobile mRNA profiles, transcriptomes, and metabolomes between rootstock (pumpkin) and scion (watermelon). The result indicated that photosynthesis-related products were regulated not only by the scion's own RNA but also by mobile mRNA from the rootstock.
However, the manuscript is completely based on the sequencing data analysis of RNA-seq and metabolomes without any verification. After careful reading and evaluation, there are major problems in the manuscript at current state. I recommend rejection of the manuscript. The specific comments are as follows:
- The article lacks innovation. Although the study compared the mobile mRNA profiles, transcriptomes, and metabolomes from the rootstock (pumpkin) and scion (watermelon), the methodology of the study lacked sufficient innovation. The findings of this paper are similar to those of the existing literature and failed to provide new insights or breakthrough discoveries.
Response: The idea of this study is relatively simple, but our data provide many mobile-RNA data of plant graft (834 mb-mRNAs and 34 and 27 different expressed mb-RNAs). It enables us to have a deeper understanding of grafting. For example, arachidonic acid metabolism pathway genes showed mobility, which have not been reported before.
- The authors' interpretation of the results in the Discussion was superficial. Several findings are presented as descriptive data without mechanistic explanations. For instance, how do photosynthesis-related mb-mRNAs specifically regulate metabolite synthesis? It is recommended to enhance the Discussion by linking these observations to signaling pathways or established models.
Response: thanks for the suggestion, we have added discussions on these contents in the discussion section. For example, we have discussed the limitations of the RNA-seq technique in identification of mobile RNAs, evidence for translation (e.g., by proteomics) might provide more compelling support for the impact of mobile RNA on improving scion quality. As showed in ‘2.6. Integrated analysis of transcript and metabolite profiles’, photosynthesis proceed by plants convert light energy into chemical energy. The carbohydrates (glucose) produced during photosynthesis can provide precursor substances for other carbohydrates. Meanwhile, these carbohydrates can be decomposed into intermediate products, which are precursor substances of amino acids, fatty acids, and secondary metabolites. We have added this information in the manuscript.
- The quality of graphs and charts needs to be improved. Using clearer labeling and color differentiation.
Response: we have changed the graphs in the manuscripts. the similar colors in the picture were modified to increase the distinctiveness.
- The description of c protocols is overly brief. Critical parameters must be added to enhance reproducibility, including: Sequencing depth, Alignment software version and specific parameter settings, Filtering thresholds for raw reads.
Response: The RNA-seq method has been described in the previous paper, resulting in an overly brief protocols of RNA extraction and sequencing. We have added some descriptions in the new submission. line 315-326
- The terminology needs to be harmonized: “mb-mRNA” and “mobile mRNA” are used interchangeably in the text, and it is suggested that the term “mb-mRNA” be harmonized throughout the text.
Responses: All the term in the text have been harmonized. we have changed the ‘mobile mRNA’ into ‘mb-mRNA’.
- The current conclusion is overly general. It is recommended to explicitly summarize key findings, such as the roles of specific genes (e.g., CmoCh20G002790[MADS-box transcription factor] or pathways (e.g., photosynthesis and carbon fixation), and to clearly outline future research directions.
Responses: We have modified the conclusion section, added specific genetic information, and prospected the future research directions.
Round 2
Reviewer 1 Report
Comments and Suggestions for Authors
I am satisfied with the outcome.
Author Response
thanksReviewer 2 Report
Comments and Suggestions for Authors
The key genes obtained from the RNA-seq were not validated by RT-qPCR.
Due to the discussion was still superficial, I recommend reconsider after major revisions.
Author Response
The key genes obtained from the RNA-seq were not validated by RT-qPCR.
Respond:
The transcriptome data were obtained from another article“Transcriptomic and metabolomic analysis reveals improved fruit quality in grafted watermelon”, in which some genes were selected for qPCR verification. In this article, we have also selected some genes for qPCR verification, and the results are presented in Supplementary Figure1.
Due to the discussion was still superficial, I recommend reconsider after major revisions.
Respond:we have added some disccusion in the new submission. line104,line311